# A systems map of the determinants of child health inequalities in England at the local level

Patricia E. Jessiman[1]*, Katie Powell[2], Philippa Williams[1], Hannah Fairbrother[3], Mary Crowder[2], Joanna G. Williams[1], Ruth Kipping[1]

1 Department of Population Health Sciences, University of Bristol, Bristol, United Kingdom, 2 School of Health and Related Research, University of Sheffield, Sheffield, United Kingdom, 3 Health Sciences School, University of Sheffield, Sheffield, United Kingdom

☯ These authors contributed equally to this work.
* Tricia.Jessiman@bristol.ac.uk

**Data Availability Statement:** The data for this study is now available in the data repository ReShare http://doi.org/10.5255/UKDA-SN-854532 https://reshare.ukdataservice.ac.uk/854532/.

## Abstract

Children and young people in the UK have worse health outcomes than in many similar western countries and child health inequalities are persistent and increasing. Systems thinking has emerged as a promising approach to addressing complex public health issues. We report on a systems approach to mapping the determinants of child health inequalities at the local level in England for young people aged 0–25, and describe the resulting map. Qualitative group concept mapping workshops were held in two contrasting English local authorities with a range of stakeholders: professionals (N = 35); children and young people (N = 33) and carers (N = 5). Initial area maps were developed, and augmented using data from qualitative interviews with professionals (N = 16). The resulting local maps were reviewed and validated by expert stakeholders in each area (N = 9; N = 35). Commonalities between two area-specific system maps (and removal of locality-specific factors) were used to develop a map that could be applied in any English local area. Two rounds of online survey (N = 21; N = 8) experts in public health, local governance and systems science refined the final system map displaying the determinants of child health inequalities. The process created a map of over 150 factors influencing inequalities in health outcomes for children aged 0–25 years at the local area level. The system map has six domains; physical environment, governance, economic, social, service, and personal. To our knowledge this is the first study taking a systems approach to addressing inequalities across all aspects of child health. The study shows how group concept mapping can support systems thinking at the local level. The resulting system map illustrates the complexity of factors influencing child health inequalities, and it may be a useful tool in demonstrating to stakeholders the importance of policies that tackle the systemic drivers of child health inequalities beyond those traditionally associated with public health.

## Introduction

Children and young people's health in the UK is in crisis, with worse outcomes than in many similar Western countries and increasing health inequalities between children and young

**Funding:** This study is funded by the National Institute for Health Research (NIHR) School for Public Health Research (Grant Reference Number PD-SPH-2015). The award went to RK, HF and KP. https://sphr.nihr.ac.uk/. The views expressed are those of the authors and not necessarily those of the National Institute of Health Research or the Department of Health and Social Care. The funders had no role in study design, data collection and analysis, decision to publish, or preparation of the manuscript.

**Competing interests:** The authors have declared that no competing interests exist.

people (CYP) from the least and most deprived areas of the UK [1,2]. The situation is exemplified by the infant mortality rate (IMR), one of the most important indicators of overall population health, which had been improving steadily in the UK up to 2014 but has since stagnated, and in England the IMR has increased [3]. This stagnation in IMR has not been seen in comparable European countries and the UK now has one of the worst IMRs in Europe [4]. Highlighting the rising inequalities, an analysis of the increase in IMR in England from 2014–2017 found that it disproportionality affected the poorest areas of the country, while many affluent areas saw the IMR unaffected [5]. This trend is echoed across many health outcomes for CYP, including mental health, obesity and oral health [6–8]. Child health outcomes also vary across ethnicity, with CYP from Black and Minority Ethnic groups consistently at a disadvantage [9]. These inequalities are persistent and increasing.

These trends are of public health concern, not only for their impact on CYP's present health but also for their future health status, with CYP who experience disadvantage in the early years having a higher risk of premature death in adulthood [10]. Over the last 20 years evidence reviews have repeatedly emphasised the need to provide better support at an early stage in children's lives if we are to have any chance of significantly reducing the inequalities in life chances experienced by people in England [11–13]. Despite this, essential services for child health are being cut, with disadvantaged areas often seeing the steepest cuts [14,15]. The health and social care reforms in 2013 transferred responsibility for public health services to local government authorities, and presented an opportunity for them to align the commissioning of preventative health services, for example health visiting, with their existing responsibilities including in spatial planning, licensing, environmental health, early years' education provision and early intervention work with children and families. However, emerging evidence suggests that while some alignment of public health priorities is being achieved at a local authority level, commissioning and decision-making relating to the health of CYP has become increasingly fragmented [16]. In England all NHS organisations are working closely with local authorities to coordinate public sector services with the perspective of working together as a system. The aspiration is that by 2021 all areas will have formed an Integrated Care System (ICS) [17]. The landscape for CYP health policy locally is complex, comprising a large number of organisations with different organisational structures, professional cultures and priorities. Such a landscape is particularly challenging when seeking to improve and reduce inequalities in children and young people's health. Child health inequalities are complex; they are created, maintained and exacerbated through multiple, related pathways [10]. One way to understand and work with this complexity and address the fragmentation of child health policy and practice is to apply a systems-based approach to identifying child health inequalities at the local level.

Systems thinking has emerged as a promising approach to addressing complex public health issues in recent years. It conceptualises poor health and health inequalities as outcomes of a multitude of interdependent elements within a connected whole (a system] [18]. Systems thinking encourages the consideration of how different actors (individuals, populations, or organisations) relate to one another and how activities in one part of a system may affect another. Applying a systems lens to thinking about public health challenges can support the evaluation of policy and programmes, and the development of interventions which recognise wider system influences [19–21]. Within public health, systems thinking has often been applied through taking a 'whole-system approach', conceptualised in a recent systematic review as those approaches that "consider the multifactorial drivers of public health or the social determinants of health, that also involve transformative co-ordinated action (including policies, strategies, practices) across a broad range of disciplines and stakeholders, including partners outside traditional health sectors" [22]. One tool to aid systems thinking is the systems map. The Foresight Obesity map [23], for example, has facilitated systems thinking and the

hypotheses of solutions to rising obesity trends which range from individual behavioural change through to population-level policy approaches [24]. System mapping draws on concept mapping techniques [25] and other qualitative research methods such as reviewing the literature, interviews with specialists, and workshops with stakeholders.

The majority of published studies using system approaches to address child health have focused on healthy eating and obesity, with a paucity of studies using systems approaches to other child health outcomes [19,21,26–29]. The current study seeks to extend the scope of a systems mapping approach beyond single child health outcomes to address inequalities in child health more broadly. In defining child health, we followed the example of the National Health Service (NHS) by including all children and young people aged 0–25 years [30]. The overall aim of the current study was to scope and create a child health system map for use at a local level in order to inform opportunities for effective interventions at a systems level to reduce child health inequalities. It sought to capitalise on the public health remit of local government in England, and the need to address persistent and increasing child health inequality through partnerships between a range of local organisations with responsibility for child health. The purpose of this paper is to describe a systems approach to mapping the determinants of child health inequalities at the local level in England for young people aged 0–25, and present and describe the resulting map.

## Method

This study used qualitative methods and we have followed the Consolidated Criteria for Reporting Qualitative Research (COREQ) checklist [31]. All participants were sent detailed information leaflets about the study and had the opportunity to ask the research team questions about participation. All provided written informed consent prior to participation in the interviews, workshops (signed consent form), or online survey (online consent form). Where participants were under 16 years of age, parents/carers were also sent study information and had the opportunity to decline consent for their child to participate (the child also had to give written consent prior to participation). The study received ethical approval from the University of Sheffield School of Health and Related Research Ethics Committee on 16th March 2019 (ref 025460).

### Research team

The research team comprised academics from public health centres at two English universities. They are experienced in the application of qualitative methods (PJ, KP, HF, MC) and public health research (RK, KP, HF, PW, JW). PJ and KP are joint senior authors of this paper.

### Study design

We used a qualitative soft systems method, based on group concept mapping [25] to develop a system map of child health inequalities at the local level. Group concept mapping is a participatory method that takes a staged approach to the development of a conceptual framework for how a group views a topic [32–34]. The approach was chosen to enable us to develop consensus regarding depiction of the child health system, which is important in the development of a shared understanding of the problem of child health inequalities among people working to address them. Initial fieldwork was undertaken in two contrasting local areas to develop locality-specific maps, which were then used by the research team to develop a single generic version that was reviewed and amended through online consultation with academics policy and practice experts. A flowchart depicting the main stages of the study methods is shown in Fig 1, and each stage of the methodology is described in detail below.

Site selection of two local authority areas (sites 1 and 2); qualitative interviews and document review in each

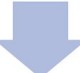

Local group mapping workshops in sites 1 and 2

Analysis and local map development; validation with local stakeholders

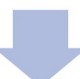

System maps for site 1 and site 2 compared and contrasted to develop a generic version of the system map

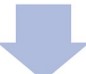

Consultation on gereic map with national stakeholders (2 rounds of electronic survey)

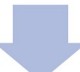

System map of the determinants of child health inequalities at the local area level

**Fig 1. Overview of methods.**

## Site selection

Two local authority areas were selected for the development of locality-specific system maps. These areas were chosen for their contrasting geography, local governance arrangements and

population demographics (to help ensure that subsequent development of a generic version of the system map was informed by contrasting locality-specific maps). Site 1 was a city in the North of England, Site 2 was a rural county in the South West. The sites also vary in governance (unitary and two-tier authority), level of deprivation according to the Indices of Multiple Deprivation for local authorities [35], and proportion of Black and Ethnic Minority residents [36].

Sites were approached early Spring 2019. Approval for participation in the study was granted by the local authority Director of Public Health in each area and by Directors in the Clinical Commissioning Group (CCG) for each area. Fieldwork was undertaken between May 2019 and February 2020.

## Qualitative interviews and document review

Face to face, in-depth qualitative interviews were held in each local area with senior decision makers with responsibility for child health. Participants were recruited through our lead contacts in the public health departments and local CCG. Participants were sent an information sheet about the study by email in advance of the interview and gave written consent to participate. A topic guide was developed for the interviews that covered the local context, key challenges, priorities and recent key initiatives in relation to child health outcomes and inequalities for CYP aged 0–25, and experience of whole-system approaches to child health (generally, and specific to the local area). The interviews were also used to identify key local documentation (policy and strategy documents), and further participants for both interviews, and involvement in mapping workshops. Interviews lasted between 30–60 minutes and were digitally recorded and transcribed verbatim. The transcripts were reviewed for key information related to the local child health context; and for sections of data that provided additional information on the local determinants of child health inequalities (described below).

Local documents (e.g. Children and Young People Plan; Health and Wellbeing Strategy; Clinical Commissioning Group strategy) were reviewed to support the researchers' understanding of the local context to enable facilitation of the mapping workshops and interpretation of results.

## Group mapping workshops

Group concept mapping has long been used in public health to address issues with a range of divergent stakeholders who have an interest in a particular topic, or are affected by an outcome [25,37]. It is valued as a tool for strategic planning, breaking down broad issues into their component parts [38] and has the advantages of eliciting and integrating tacit knowledge from a broad range of stakeholder-types (including policy makers, practitioners and commissioners), can be used with the community to support their involvement in decision-making, and can act as a basis for practical decision making in a public health context [39].

Group concept mapping typically involves the following steps: agreeing the focus and the primary question(s) of interest; individual brainstorming and generation of statements; group structuring of statements; rating and sorting using multidimensional scaling and cluster analysis; and final development of the map [33]. However the method can be successfully adapted from the original approach to include data generated through face to face brainstorming, qualitative interviews, and published literature; adapting or omitting the rating phase, and use of surveys and interviews to gain structured feedback. These adaptions allow for greater accommodation of diverse perspectives, the inclusion of data from multiple modalities, and additional input from experts, end users and the public during more than one step in the traditional group concept mapping approach [37]. Accordingly, we adapted Kane and

Trochim's [25] group concept mapping approach by supplementing the early stages with qualitative data from interviews with expert stakeholders, and a documentary review of key policies and strategies related to child health in each local site. These informed recruitment to, and facilitation of, group mapping workshops in each site. We structured the maps using qualitative thematic analysis, informed by an established framework from published literature. The initial structures of the local site maps were refined by local stakeholders at later meetings.

Group mapping workshops were held in each site with different stakeholder groups: professional staff from the local authority, NHS and CCG, third sector organisations, and elected members with responsibility and expertise in child health; children and young people recruited through local youth voice and empowerment organisations; and parents/carers. A framework and guidance document for the group concept mapping workshops was developed by PW and PJ and included guidance on workshop facilitation (informed by Hovmand et al's group model building facilitation handbook [40]); pre-workshop preparation including recruitment and information for participants; and workshop tasks and timings. A separate workshop guidance document for mapping workshops with children and young people was developed by KP and MC in consultation with our community advisor, and tested and refined through public involvement and engagement activities with children and young people.

One week prior to the professionals' workshops, participants were sent an email with basic information about whole systems approaches in public health, and asked to consider the following two questions:

a. *What factors drive child health and wellbeing in [Site name]?*

b. *What factors drive health inequalities between different groups of children and young people in [Site name]?*

The process for CYP workshops differed slightly, in that questions were reworded to be more age-appropriate and were not shared in advance with participants. Instead CYP workshops included an initial 'warm-up question' (*"What does good health and well-being mean to you")* to encourage participants to think broadly about the meaning of health and well-being before addressing the main workshop questions: '*What influences good health and wellbeing for children and young people in [Site name]?'*; and *'Why do some children and young people in [Site name] have better health and wellbeing than others?'*

Participants were encouraged to consider as a factor anything that could be a determinant of child health outcomes. Examples provided to participants included services (health, education, transport etc.; local environmental factors (green space, air quality); behaviours (activity level, diet), and influences (peers, family, community etc), but participants were not constrained to these and encouraged to think broadly at this stage. Workshops lasted for between one to two hours and involved the following steps:

1. An introduction to the research team members present; overview of the workshop including timings and collecting consent forms.

2. A short presentation led by the research team that included an introduction to systems thinking and its application in public health, and the group concept mapping process.

3. Generating influencing factors: Individual generation of factors in response to questions a) and b) above. Participants worked individually, and were encouraged to write each factor on a blank index card. Facilitator prompts during this task included:

   - *Generate as many as factors as you can think of*

- *Factors should be within the control or influence of agencies at the local level (e.g. the local authority, NHS, CCG, third sector organisations)*

- *Think specifically about [Site name]; not any local area*

- *There are no bad ideas, write down anything you feel is important*

- *The factors need to be dynamic (variables that are capable of change)*

- *Consider **all** aspects of health*

- *Include **all** aspects of inequalities*

- *Consider all children and young people in (Site name) aged 0–25 years*

4. Sharing and clustering: Participants worked in small groups facilitated by one of the researchers. They were invited in turn to share a single factor and explain why they had written it and its relationship to questions a) and b) above. Participants were also invited to consider the relationship with other factors already discussed. For each factor, facilitators prompted participants to be clear on *i) what aspect of health is affected? How? Why? ii) what are the relationships with other factors on the table and iii) What element of inequality is affected?* At the end of each explanation the participant was invited to place the index card on the table, and if appropriate, place the card next to other factors they considered closely related. Participants took turns to share factors until they were all complete (or time ran out). They were encouraged to discard index cards which duplicated factors already discussed, and to write down new factors that occurred to them during this sharing and clustering process. This stage was the longest and took the majority of workshop time.

5. Sharing discussions: Where time allowed, group facilitators gave a brief summary of the factors on the table and any clusters or groupings that emerged, the main issues relating to inequalities, and how much consensus or differences in perspective there was in the group discussion.

The group discussions were audio recorded and augmented with detailed notes made during the sharing and clustering stage by a researcher not involved in facilitating the groups, to ensure the narrative around each factor was captured, including links to child health outcomes, inequalities, and to other factors already on the table.

### Analysis and local map development

Following the initial group mapping workshop with professionals in each site, the index cards with factors, group transcripts and detailed notes of the narrative around each one as described by participants were carefully tabulated by at least two members of the research team (MC, KP and HF Site 1; PJ and PW in Site 2). We used a systematic method to tabulate, for each factor, the factor name (copied verbatim at this stage), its effect(s) on a child health outcome or behaviour, links or relationship with other factors, and a summary of the narrative, all as described by workshop participants. In this initial stage, factors were frequently linked by participants to several effects and other factors, all of which were captured and tabulated. Examples are shown in Table 1 below.

This process was continued for all factors raised by participants during the group mapping workshops. As participants worked in small groups during the sharing and clustering stage, there was some inevitable duplication of factors. Each duplicate -pair was compared to review the narrative, effect(s) and link(s) to other factors to ensure that no data was lost as duplicate factors were removed. The same process was followed for workshops held with other stakeholder groups in each site (children and young people, and carers).

**Table 1. Examples of factors provided by workshop participants.**

| Factor name (as written on card) | Narrative (from workshop participants) | Effect | Links to other factors |
|---|---|---|---|
| Availability of calorie-dense food | *In the most deprived communities there is more likely to be a higher number of fast food joints–a neighbourhood with high deprivation is less likely to have healthy food choices. Leads to poor diet and obesity.* | Poor diet Obesity | Neighbourhood deprivation Number of fast food outlets in residential areas Availability of healthy food |
| Education about cleaning teeth | *Poor education about the importance of cleaning teeth leads to poor dental carer and oral health. This is also not helped by limited access to NHS dentists in some areas, and very limited access to orthodontists. Young people end up with poor teeth, which is embarrassing and upsetting* | Oral health Emotional wellbeing | Access to NHS dentists Access to orthodontal care |
| Housing quotas | *The district council has a lot pressure to meet UK government housing quotas, which results in limited housing types. Land for building is difficult to acquire, so builders want to maximise return. We have a lot of a very specific type of housing here, is crammed in, usually very small—lack of local amenities and recreational and around it* | | Housing quality Limited access to amenities |

Data from the workshops was augmented with data from the initial interviews with senior decision makers with responsibility for child health, and additional factors and relationships were identified in the transcripts of these interviews.

In each of the two sites, once the data from group mapping workshop, and qualitative interviews had been tabulated, further thematic analysis of the factor lists with their associated narrative and links was undertaken to consolidate factors. Guided by a commitment to privileging our participants' perspectives, we sought to ensure our analysis was data-driven (an inductive approach). However, following a review of existing frameworks of the determinants of child health inequalities and expert consultation, we used Goldfeld et al's conceptual model of neighbourhood effects influencing early childhood development to organise the factors identified by participants and 'clustered' them into meaningful sub-systems or 'domains' [41]. Our approach therefore cohered with an appreciation that thematic analysis is 'actively constructed' by the researcher and that 'analysis lays over bits of data to give them shape without doing violence to them' [42]. Goldfeld's conceptual model offered the 'best fit' for working with our data as the domains echoed our data to a large extent.

Factors were entered as nodes on the mapping software VUE, a free concept and content mapping tool developed by Tufts University [43], and links drawn where indicated by the data analysis. An initial, locality-specific system map of the determinants of child health inequalities was developed for each site.

## Validation with professional staff in local areas

These initial locality-specific maps were presented back to professional staff in each site at a subsequent meeting. During these meetings, the researchers explained the map structure (domains) and consolidation of factors. Participants were invited to review the map and comment on factor names, whether they had been grouped in the correct domain, and any missing factors within each domain. Participants could also amend links between factors, and comment on the map as a whole. Detailed notes were taken from each group, and the map refined by the research team to reflect the comments.

## Development of a generic version from site-specific maps

At the end of the stages described above we had final versions of two locality-specific system maps of the determinants of child health inequalities at the local area level. The two maps were

compared by researchers who had led fieldwork and map development in each site (PJ, MC, PW and KP) to determine commonalities and discrepancies between factors. The aim of this was to remove any area-specific factors and combine the remaining factors to develop a more 'generic' version of the child health system map that may be applied to *any* English local authority area. This was done alongside the tables of factors, narratives and links described above. A comparison of the two area maps revealed four categories:

Same: where factors had the same name and meaning (derived from the narrative), and were placed within the same domain on both site maps.

Similar-Extension: where factors fell into the same domain, but where one site had consolidated factors differently compared to the other. This meant that one map showed more detail expressed as a greater number of factors, than the other. Most factors across the two maps fell into this category.

Different: The factor only appeared in one site.

Conflict: The factor appeared in both sites, but had been assigned to different domains.

Where factors fell into the similar-extension category, in most cases the decision was taken to use the greater number of factors so that more detail was shown on the generic map. Factors under the 'different' category were examined to determine whether they were site-specific (i.e. factors driven by very localised circumstances and unlikely to be generalisable to other areas) or simply ones that had not emerged during data collection in one of the sites but may likely apply. Only the latter were included in the generic map. Where factors had been assigned to different domains, the team reviewed the narrative and links associated with it to determine the 'best fitting' domain for the generic map. Once the final list of factors had been agreed, links and effects for all were consolidated from the area-specific maps and an initial version of the generic map developed in VUE.

## Online survey

The first version of this generic map was shared with national experts in child health, health inequalities and system mapping through two rounds of an online survey using the Jisc Online Surveys platform [44]. The survey questions were developed by the research team and piloted with academic colleagues (the questions were qualitative). Seventy-two participants were directly invited by email to take part, and encouraged to forward survey details to colleagues not on the original invitation. Invited participants included experts in child health, health inequalities, and system mapping from academia; national leaders and decision makers in child health from Public Health England, the Department of Health, the NHS, the Local Government Association, and leading children's charities; local policy makers commissioners and practice collaborators including Directors of Public Health, and practice collaborators working in child health settings.

The first survey was open between December 2019 and January 2020 and provided an overview of systems thinking in public health and the study aims, and explained the methods used to develop the generic map. Participants were shown the whole map and invited to comment on individual domains. For each domain, participants were asked a) whether any factors were missing b) if so, how these new factors linked to others already on the map and c) if any factors or links were incorrect. Based on the feedback received, further refinements were made to the generic map and a second version sent to the same group of participants in a second online survey in January and February 2020. Changes to the first version were outlined in detail and respondents asked to indicate their agreement (or not). In addition, participants were asked a

series of open questions about the utility of the map, including its capacity to represent the needs of subpopulations of children and young people; its potential use at the local level, including informing discourse and identifying potential strategies to address child health inequalities.

Details of both surveys are in included in S1 Appendix.

## Results

### Participants

**Local area qualitative interviews.** A total of 16 professional staff across the two local sites participated in qualitative interviews prior to mapping workshops. In site 1, eight interviews were conducted with professionals, including one from the NHS (acute hospital Chief Executive) and seven from the local authority (Director of Public Health, two Health Improvement Principals, three Heads of Commissioning and one Commissioning Manager). In Site 2, eight participants were recruited, including three from the local authority (Deputy Director of Children's Services, Consultant in Public Health, and the Early years and Primary Adviser); three from the Clinical Commissioning Group (senior managers and Directors with responsibility for quality, nursing, mental health, learning disabilities and women and children's health) as well as a Police Superintendent (also a member of the local Child Partnership Board), and the Chief Executive of a third sector organisation delivering mental health services to CYP across the county.

**Group concept mapping workshops.** Three mapping workshops were held in each site, with 73 participants in total. In Site 1, one workshop was held with professional staff and two with children and young people; in Site 2, one workshop was held with professional staff, one with children and young people, and one with parents/carers. A diverse sample of senior decision makers with responsibility for child health from across the local authority (LA), Clinical Commissioning Group (CCG) and NHS, third sector organisations, police and fire service, schools, and elected officials was achieved in both sites. The sample of CYP and carer participants achieved across the two sites represented children of different ages and ethnic backgrounds. Details of participants in each workshop are shown in Table 2.

**Table 2. Workshop participants.**

| Group concept mapping workshops -participant type | Site 1 | Site 2 |
|---|---|---|
| **Professional staff (N = 35)** | N = 22 | N = 13 |
| | (Includes professional staff from the local authority, CCG, NHS, elected councillors, third sector and police) | (Includes professional staff from the local authority, CCG, NHS, elected councillors, third sector, and fire service) |
| **Children and Young People (N = 33)** | N = 19 (across 2 workshops) | N = 14 |
| | (Age range 12–24 years; 7 female, 1 non-binary; 8 from Black and Minority ethnicity background (1 undeclared ethnicity) | (Age range 10–20 years; 9 female; 3 from Black and Minority ethnicity background) |
| **Parents/Carers N = 5** | - | N = 5 |
| | | (3 female; all white British; All parent/carers of children with special educational needs and disabilities aged between 7–22 years) |
| **Total participants** | N = 41 | N = 32 |

**Validation with professional staff in local areas.** Initial maps were presented back to professional staff in each site. In Site 1, this happened over a series of four meetings, the first a large group of stakeholders, most of whom had been involved in map development (N = 26) and the later three with a smaller working group (N = 6). At each meeting the latest version of the local map was presented for discussion on its accuracy and potential local utility. In site 2, a single validation meeting was held with nine professional staff who had been involved in map development.

**Online survey.** There were 21 participants in the first round of the online survey. Nine of these were public health academics from English universities (one was also a General Practitioner); 5 from directors or consultants in public health working in English local authorities; 4 national and regional programme leaders from Public Health England (an executive agency of the Department of Health and Social Care with responsibility for population health and well-being, and reducing health inequalities); 2 commissioning leads for children and maternity from CCGs; and one from a director of an independent third sector research organisation.

The second round of the survey had eight respondents, three of whom had also responded to round one. One was a public health academic; 3 directors or consultants in public health working in English local authorities; 1 programme leader from Public Health England; 2 commissioning leads for children and maternity from CCGs; and a community paediatrician.

## The system map of the determinants of child health inequalities at the local area level in England

The final map of the determinants of child health inequalities at the local area level, developed through comparison of the two final locality-specific maps and online consultation with national experts, has 125 factors arranged across six domains (see S2 Appendix). The map also shows over 300 links between individual factors that have been made by participants in local areas or online consultations. The links are made both within and across domains, indicating the complex interaction of factors across the map. The use of domains is intended to support understanding of the whole system, and the cross-domain links between factors indicate the interdependency between those domains.

Goldfield et al's conceptual model of neighbourhood effects influencing early childhood development, used to organise the factors identified by participants, has five interconnected domains: *physical*, *social*, *service*, *socio-economic*, *and governance* [41]. Following analysis of our data from the two sites, we adapted this model by amending the *socio-economic* domain to just economic, and moving socio-demographic factors into the social domain to capture the separate influences of the social world (family, peers and community) within one domain. In addition, a number of factors did not fit into any of these five domain areas. Some of these referred to children and young people's outcomes and behaviours that may be influenced by factors in other domains, but also may interact and influence each other; a sixth, *personal* domain, was added to capture these. The use of this amended framework was accepted by participants during validation meetings in both sites, and by participants in the online consultation of the generic version of the map.

This is a qualitative map. Factors are not weighted by impact on child health inequalities, and links between them are non-directional and do not attempt to capture causality (i.e. we do not attempt to indicate whether an increase or decrease in factor *A* would result in an increase/decrease in any linked factor *B*, only that some change will occur). This is due to the nature of the raw data used to develop the map; while participants in mapping workshops were asked to indicate links between factors they mostly did so without indicating direction of

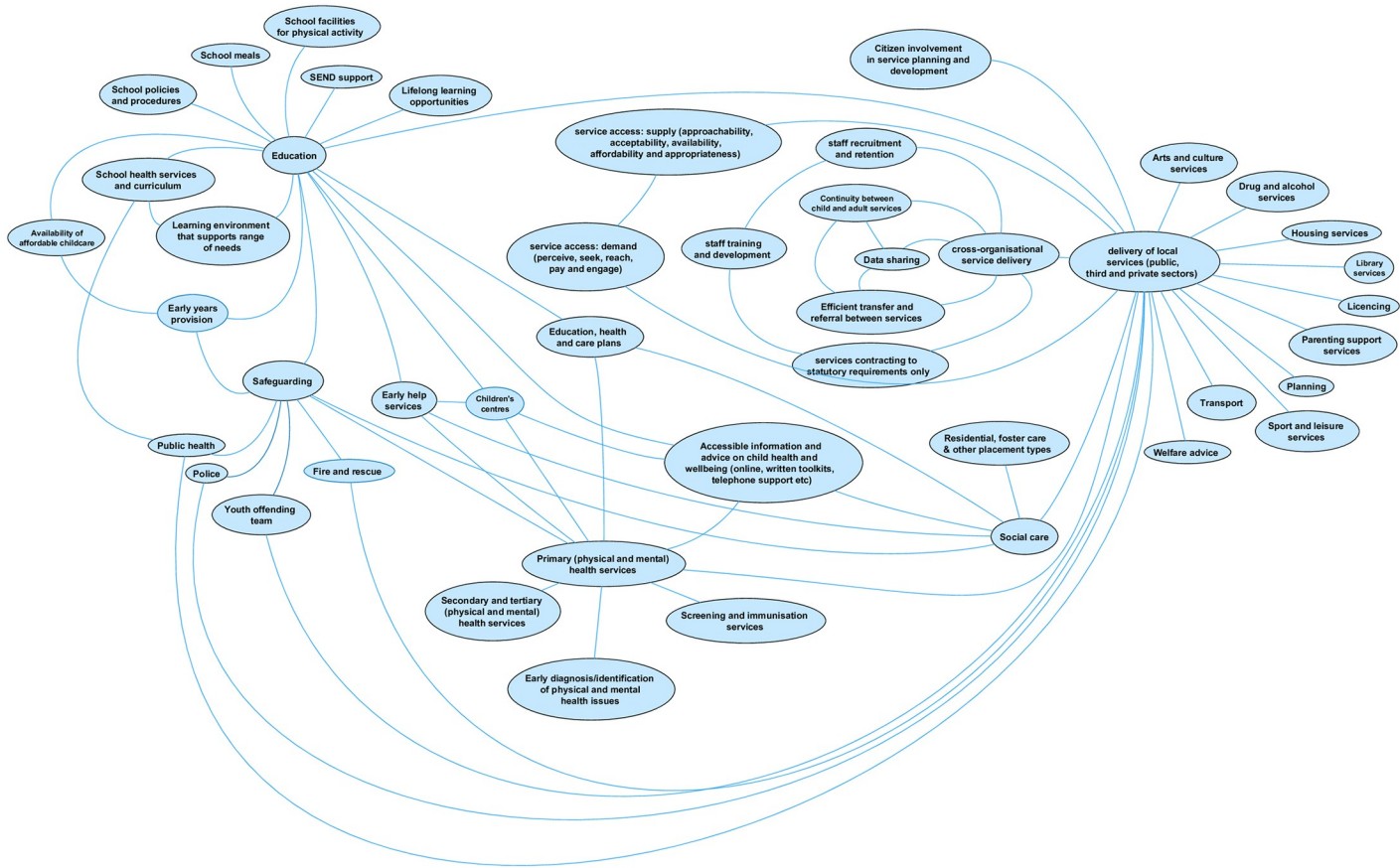

**Fig 2. The service domain.**

effects (unless explicitly prompted) and we did not have sufficient data to add directionality to the links.

**The service domain.** The service domain (Fig 2) comprises 46 factors and represents the availability and delivery of services (provided by the public, private and third sectors) at the local area level that impact on child health outcomes. The raw data that emerged from group concept mapping workshops produced numerous factors influencing the quality, quantity, accessibility and/or coordination of individual service types. These were consolidated during the analysis into factors including *staff training and recruitment*, *data sharing* and *efficient transfer and referral between services* (including between child and adult services), *citizen involvement*, and *service accessibility* (see Fig 3 below) and linked to the delivery of *all* local services. Service accessibility was a particularly strong concern. In Site 2, this was often in relation to the rurality of much of the county and the physical distance to services; concerns about accessibility for differencing subpopulations of children and young people emerged in both sites. To capture over 30 initial factors that emerged, we applied the conceptualisation of access to health care developed by Levesque *et al* [45] to describe both demand- and supply-side factors of accessibility, again adapting the framework to apply to all services. Supply-side dimensions of accessibility include *approachability* (transparency, outreach, information and screening); *acceptability* (professional values, norms, culture and gender); *availability and accommodation* (location, accommodation, opening hours, appointment mechanisms); *affordability* (direct, indirect and opportunity costs) and *appropriateness (*quality, adequacy,

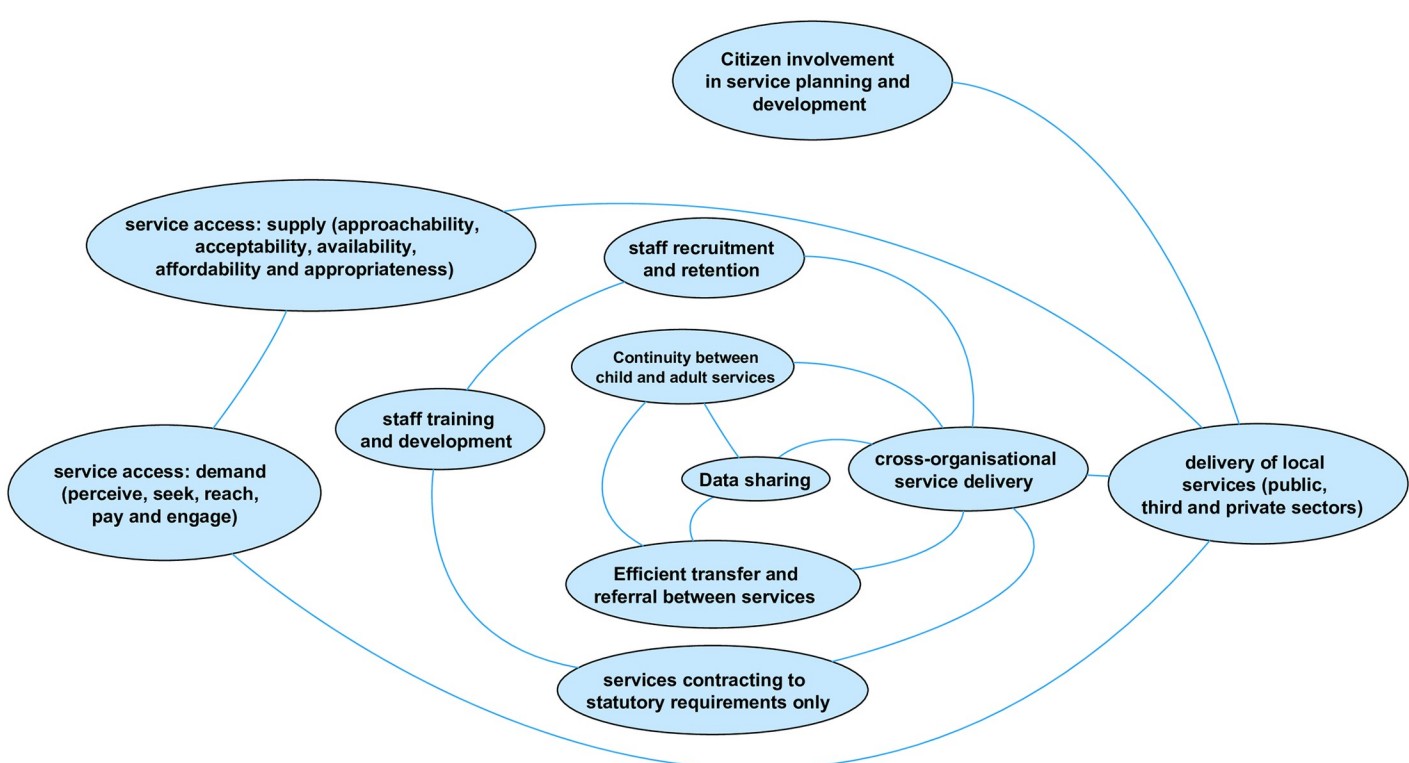

**Fig 3. Factors influencing the quality, quantity, accessibility and/or coordination of all services.**

coordination and continuity). Demand-side dimensions of accessibility include *ability to perceive* (literacy, beliefs, trust and expectations); *ability to seek* (personal and social values; culture, gender, autonomy); *ability to reach* (living environment, transport, mobility, social support); *ability to pay;* and *ability to engage* (information, adherence, caregiver support).

The three factors with most links to those in other domains are unsurprisingly health, social care, and education. The decision to represent *all* health services by only three separate factors on the map (primary, secondary and tertiary, and public health) was often commented on by respondents to the online survey. Several respondents wanted to add more specific health service provision into this domain, in particular health visitors and school nurses (perhaps reflecting the number of respondents working in public health). This was a deliberate decision by the research team, in order to balance specificity with usability of an already complex visualisation. It will likely also help with generalisability to local areas given that service provision varies by locality. Further, health, education and social care have joint responsibility (with other services) for provision that also emerged as determinants of child health, including safeguarding, early help services, education, health and care plans (for children and young people with special educational needs and disabilities (SEND)). However, the range of service types emerging from the workshops was much wider than these core three, indicating the wide array of service provision in the local area perceived to impact on child health outcomes (and inequality of outcomes). Service types were also added as a result of the online consultation (e.g. *library services*, and *children's centres*).

**The economic domain.** The economic domain (Fig 4) comprises 12 factors associated with the economic resources available to children and young people, at both the household- and local area-level. Given what is known about the associations between household income

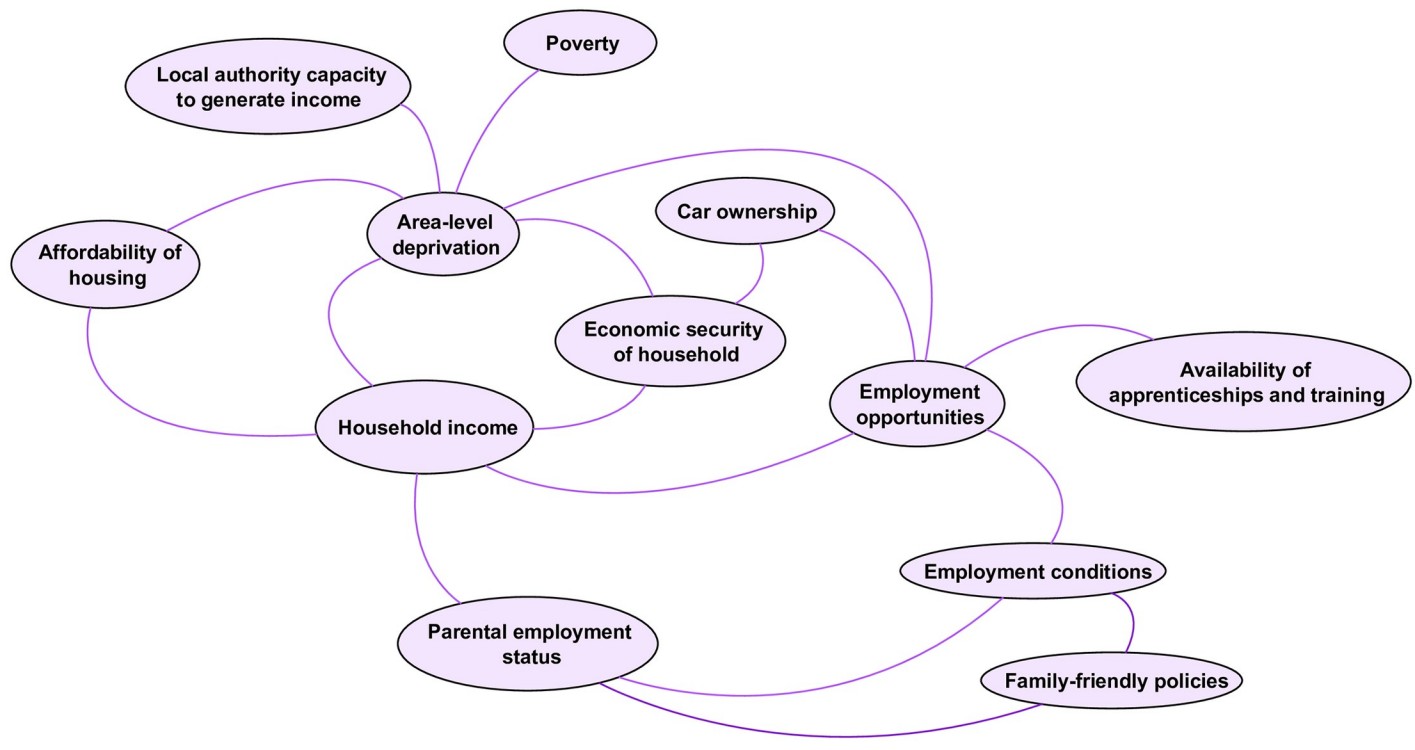

**Fig 4. The economic domain.**

[46] and area deprivation [7,12] with child health inequalities, it is unsurprising that workshop participants identified *area level deprivation* and *household income* as the two factors in the economic domain with the most links to factors in other domains, particularly health behaviours and outcomes in the personal domain. *Household income* is linked with *parental employment status*, but these are themselves influenced by area-level factors including *employment opportunities and conditions*. *Local authority capacity to generate income* relates to the powers local authorities have to raise additional revenue beyond national Government grants (national funding grants are considered outside this system given the limited capacity for change or influence at the local level). It includes measures local areas can take to raise revenue including the Adult Social Care levy on council tax; the Community Infrastructure Levy; rents, fees and charges, sales, investments and other contributions.

**The governance domain.** The governance domain (Fig 5) is concerned with the conditions associated with effective development and implementation of policy to support child health and reduce inequalities at the local area level. Participants identified that local governance and leadership will be influenced by factors outside the boundaries of the local system, in particular by policy and statutory requirements set by national government. Factors emerged that are associated with people involved in shaping local policy; the prioritisation of children and health in policy, and alignment of policy across local agencies. Key stakeholders include locally elected councillors (and in Site 2, cooperation across members elected to two tiers of local governance was especially salient); those responsible for local leadership of child health policy (and accountability for this was added during the online survey) and citizens (including children and young people). The prioritisation of children's health in local areas is linked with this leadership, as is allocation of local funds. The array of services identified as

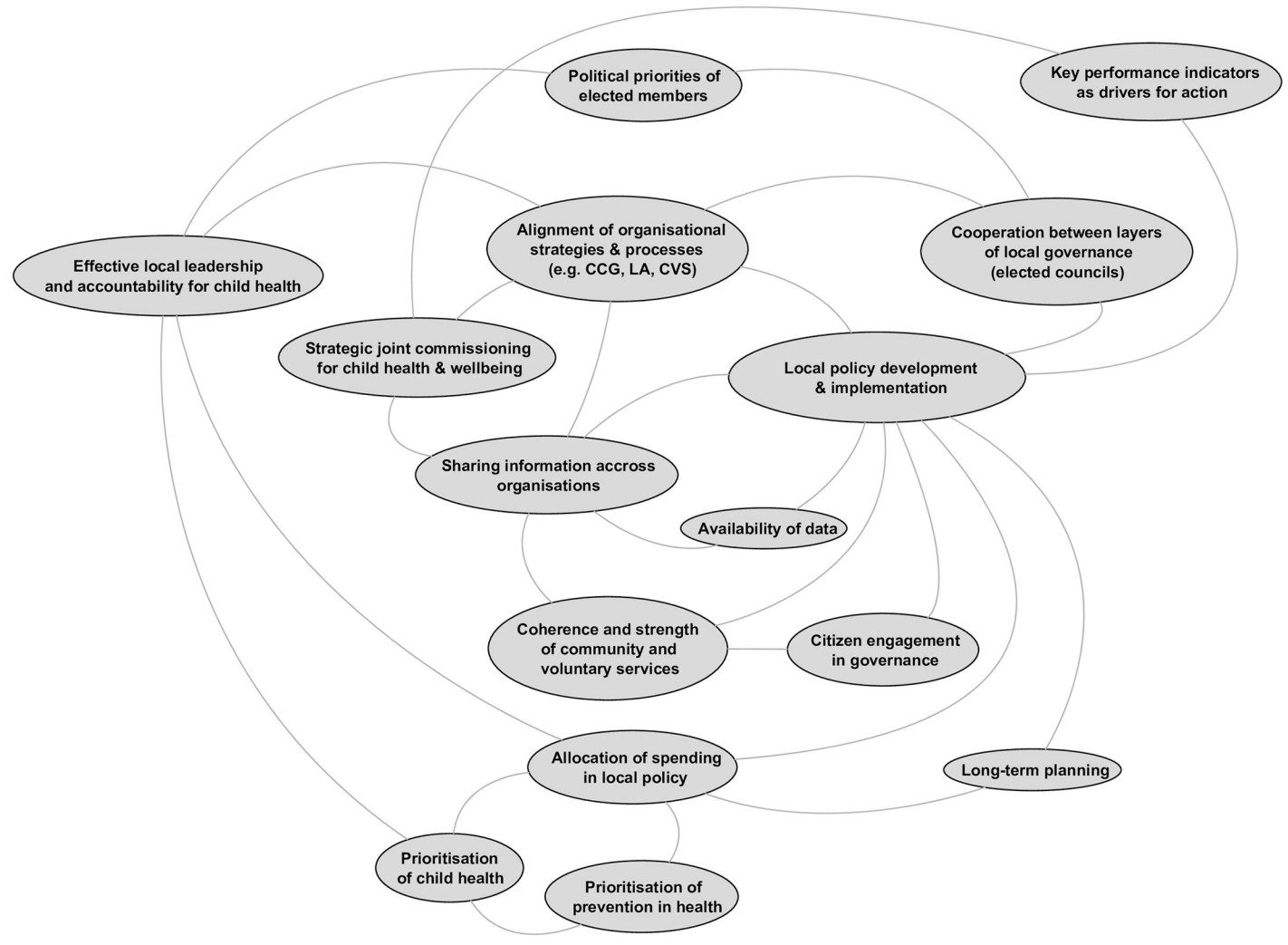

**Fig 5. The governance domain.**

important to child health outcomes in the service domain is reflected in the governance domain by the factors associated with the alignment of policy and strategy across all responsible local agencies, including those in the public, private, and third sectors (referred to here as the community and voluntary sector). This is supported by the availability, and sharing, of data to support this. The key factor at the heart of the governance domain, *local policy development and implementation*, is linked to *delivery of local services* in the service domain.

**The physical domain.** The physical domain (Fig 6) includes factors related to the physical environment in which children and young people live which are broadly split across those related to safe spaces for social activity, housing, transport and physical infrastructure, and local amenities. *Safety* emerged as a key quality for many factors in this domain, and what is 'safe' will differ across subpopulations of young people (e.g. different age groups; children with disabilities). Workshop participants emphasised the importance of local access for play, social and physical activity, and green space for children and young people's wellbeing. Online participants added arts, cultural and religious activities. A*ccessible public transport*, and *safe active travel routes*, are necessary to support CYP access to these spaces. Infrastructure to support

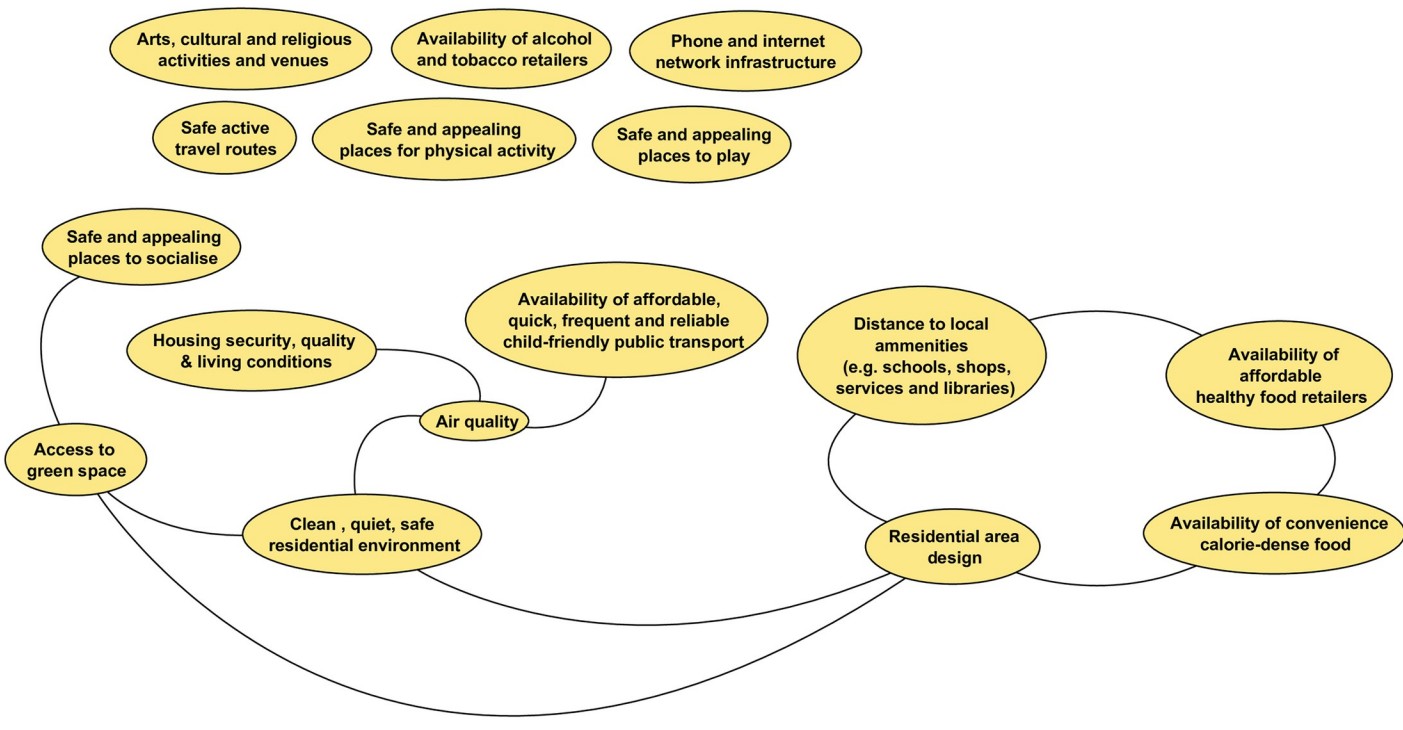

**Fig 6. The physical domain.**

mobile telephone and internet access was perceived as crucial to support CYP access to online social networking and, for residents of all ages, information and communication. Factors concerned with living conditions, not least secure, quality homes and *residential area design* that supports a clean, safe living environment, are also prominent. Finally, there are a group of factors concerned with retailers of healthy (and unhealthy) consumables including healthy food, fast food, alcohol and tobacco. Unlike Goldfield et al's conceptual model, levels of crime and disorder are not included within this domain. Factors related to these were moved into the social domain in response to feedback from the online survey that suggested these belonged alongside community-level social influences.

**The social domain.**   Factors in the social domain (Fig 7) are related to the people around CYP who influence their health behaviours and outcomes, including parents and families, peer groups, and local communities. It also includes the influences of people CYP encounter online and in the media, including social media. Most factors emerging from the workshops in this domain are related to families, in particular parents and caregivers. There were over 40 factors that related to parent's status, including their health, education, employment, culture and ethnicity, beliefs and values, drug and alcohol use, disability, and more. We have consolidated these into the factor *family/parental capacity to support the child*. Other important factors include *family stress* (linked to presence of abuse, wider family support and availability of informal childcare), *parental engagement with services*, and more simply, the availability of a *loving family environment*. At the community level, *local area demographics*, *community cohesion*, *social capital*, and *rates of crime and disorder* are included. Factors involving peers include *online and offline friendship groups*, *memberships of clubs, youth groups and societies*, and *bullying and/or peer pressure*.

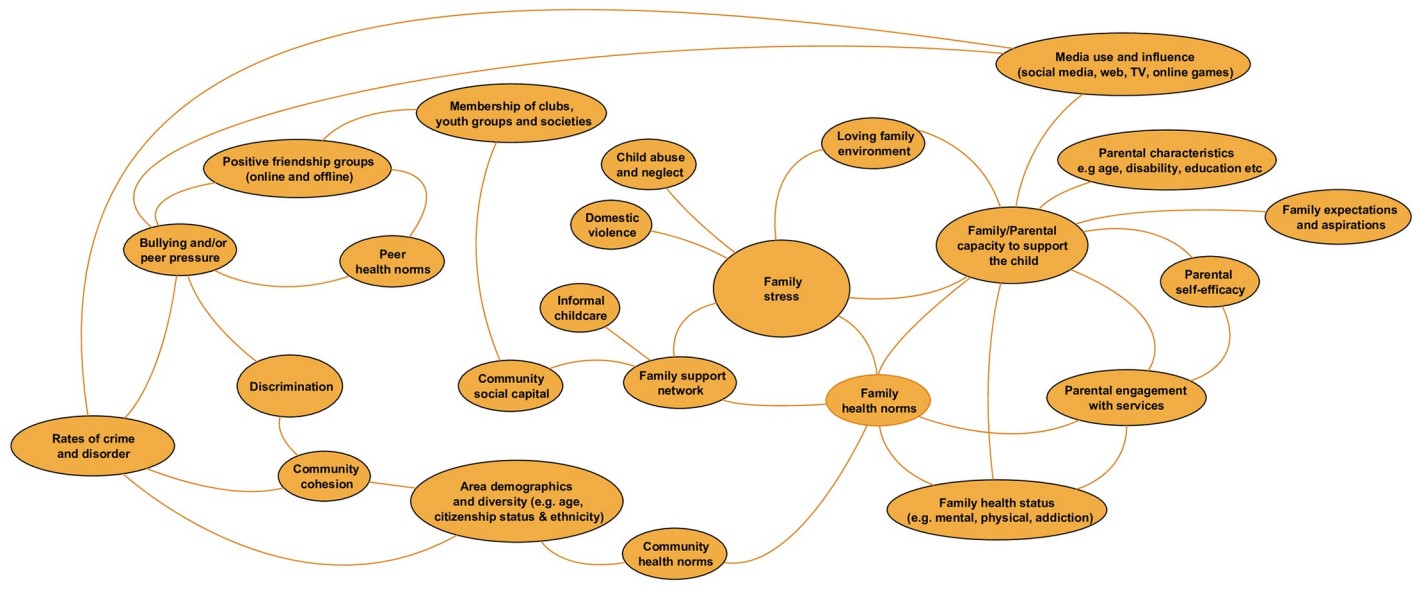

**Fig 7. The social domain.**

Health norms were also perceived as key influencing factors on children and young people's behaviours and outcomes and these are included as separate factors (or domains of influence) for peers, families, and communities.

**The personal domain.** The personal domain was added to the Goldfield et al model to capture the factors emerging from workshops that refer to CYP behaviours and outcomes. The personal domain factors are included on the whole system map (in green), and as most links to them are cross domain, they are listed in Table 3 (although some inter-domain links were identified and shown on the whole map).

Some factors in the personal domain *are* health outcomes e.g. *mental health;* some are behavioural outcomes with implications for health *e.g. diet, alcohol use.* Other outcome-types are developmental and/or educational e.g. *achievement/employability, speech and language skill,* reflecting known links between these and later life health inequalities [12].

The factor '*capacity to get from A to B*' is included as participants in both sites indicated this was key to accessing services, social spaces, and other amenities. For both young children

**Table 3. Personal domain factors.**

| | | |
|---|---|---|
| Safe-sex/contraception use | Cooking skills/food knowledge | Criminal and/or antisocial behaviour |
| Alcohol use | Diet | Risk of victimisation/exploitation |
| Smoking | Physical activity | Risk of accident/injury |
| Gambling | Oral hygiene | Exclusion/attendance rates (school) |
| Substance use | Medication use | Educational achievement/employability |
| Mental health | Vaccination status | Speech and language skill |
| Resilience | Sleep | |
| Exam-related stress | CYP capacity to get from A-B | |
| Self-esteem | Engagement with health services | |
| Self harm | | |
| Social contact/isolation | | |

reliant on caregivers, or older more independent young people, public transport, active travel routes, and car ownership are the main influencing factors on this.

Online survey respondents noted that many of the factors in the personal domain are outside the immediate influence of agencies at the local level. However we include them in the system map to illustrate how they have been linked to i*nfluencing* factors on the map by respondents in the two local authorities. For example, *self-esteem* in the personal domain has been linked to *school learning environments*, and *CYP engagement in governance*. Just as *parental engagement with services* has been included in the social domain, online respondents also added that, for older children, their own engagement with services was a key determinant of health outcomes.

**Influences outside the boundaries of the local area system.** Two further clusters of factors emerged during the analysis of local site data that were considered important determinants of inequitable health outcomes by participants, but were not amenable to local influence and therefore outside the boundaries of the system. The first of these referred to sub-populations of children and young people defined by personal characteristics and/or circumstances not amenable to local influence (e.g. gender, ethnicity, children in state care) but which heightened their risk of poor health and wellbeing outcomes. We consulted with participants present at the validation and feedback workshops in each local site on how best to represent these groups. Participants suggested that the map should be viewed through an 'inequality lens' according to these different population groups of children and young people. As an example, factors in the service domain of the map concerned with service accessibility will have varying importance for CYP of different genders and (dis)abilities. The final list of subpopulations, including amendments and additions from online survey respondents, is shown in Table 4 and on the final map as a means of encouraging local decision-makers to review the differential effects of the child health system on these subpopulation groups. This list is similar to that presented in recent work by the Children's Commissioner for England on identifying and mapping vulnerable groups of children in England [47].

The second emergent cluster referred to societal or global influences, such as climate change, national government policies (in particular austerity-related measures including welfare reform and cuts to local government grants), which are not amenable to change at the local level. Local site participants felt strongly that these should be included on the local system maps to reflect the context in which local agencies are working. There was disagreement between the two sites about how best to do this, in particular on whether or not links should be present, to show how societal or global influences impact on factors that are also within the local area system (e.g. national government grants to local authorities will influence *local decision-making* and *allocation of spend*; austerity measures and welfare reform will influence *household income*). Consultation with respondents to the online survey also revealed a mixed view on this. Disagreements about the boundaries of a system, and whether and how factors should be included, are common in system approaches [48,49]. We have not included the links to factors within the local system, but listed them alongside the map as "determinants of child health inequalities outside of the immediate influence of agencies at the local level" (see Table 4).

## Discussion

This paper presents the results of a qualitative study conducted with a wide range of stakeholders (children and young people, carers, professional staff from the local authority, NHS and CCG, third sector organisations, and elected members with responsibility and expertise in child health) from two contrasting English local areas. A group concept mapping approach has

**Table 4. CYP subpopulations and determinants impacting outside influence of local level.**

| CYP personal characteristics and circumstances (subpopulations vulnerable to poor health outcomes) | Determinants of child health inequalities outside of the immediate influence of agencies at the local level |
|---|---|
| Age | Austerity |
| Ethnicity | Climate change |
| Cultural and religious background | Innovation and technological advancement |
| Low birth weight | National government funding formulae |
| Gender identity | National policy and statutory guidance |
| Sexual identity | National social, political, cultural environment |
| CYP from military families | Rural/urban geography of local area |
| Young/teenage parents | Societal attitude to children and young people |
| Young carers | Statutory constraints |
| Care leavers | Statutory inspection regime(s) |
| Young offenders | Welfare reform |
| CYP in care (including those placed out of the local authority area) | |
| CYP at risk (e.g. on child at risk registers) | |
| CYP experiencing transition (of any type) | |
| CYP living with previously unsurvivable illness | |
| CYP with physical disability | |
| CYP with intellectual disability | |
| CYP with autism spectrum disorder | |
| CYP with special educational needs and disability (SEND) | |
| CYP with experience of adverse childhood events (ACEs) | |
| CYP with complex and/or long-term physical health conditions | |
| CYP with complex and/or long-term mental health conditions | |
| CYP with genetic predisposition to disorder(s) | |
| Asylum seeker/refugee status/citizenship status | |
| CYP who are members of Gypsy, Roma and Traveller communities | |

been successfully applied to construct a system map of the determinants of child health inequalities in each area. Factors within the maps have been thematically organised into six domains by adapting Goldfield et al's [41] conceptual model of neighbourhood effects influencing early childhood development, and accepted as a valid representation of the local child health system by professional staff in each local area. These maps have been systematically compared and contrasted, and amended through online consultation with national experts in child health, to develop a 'generic' version of a visual map of the determinants of child health inequalities that may be applicable to any English local area.

Systems approaches have been frequently applied to complex public health problems, and in child health the focus has most often been on single outcomes, in particular healthy eating [27] and obesity [50]. We believe the approach described in this paper is novel in scope, by asking participants to consider inequalities across *all* health outcomes, and for all populations of children and young people aged 0–25 years. The resulting map is complex, with 125 factors arranged into six domains. The domains represent the main spheres of influence on child health, and inequitable outcomes, as perceived by participants. They include local governance,

in particular effective development and implementation of policy to support child health and reduce inequalities; the physical environment in which children and young people live; the social influences on health stemming from their peers, family, and local community; the services available to support them (including the quality, availability and delivery of a wide range of services provided by the public, private and third sectors at the local area level); local economic conditions both at neighbourhood and household level; and individual characteristics and behaviours of children and young people themselves that influence health inequalities. Links between the factors show the complexity of relationships both within, and across, these six domains. By encouraging participants to think broadly, we believe this visual map begins to encapsulate how child health inequalities may arise from the complex interaction of factors across the whole local area system and not just those factors related to one aspect such as the food system or local public services.

A key strength of our study includes the engagement of a breadth of stakeholders at the local and national level. Locally, our amended group concept mapping approach facilitated the successful engagement of a diverse sample of senior decision makers with responsibility for child health and wellbeing, including across local authority departments, CCG commissioners, and other public bodies. The workshop was adapted to facilitate the engagement of community members as a result the map also reflects the child health system as perceived by children and young people, and carers. This study adds to the growing literature demonstrating that communities can be successfully engaged in systems thinking [27], and can use it to generate a conceptual system map that captures their perceptions of a complex public health issue [48,51,52]. The twenty-six respondents across two rounds of online survey questions about the generic version of the map were also drawn from a diverse range of national expertise, including across academia, local authority public health teams, CCGs, Public Health England officials and health practitioners, all with expertise in child health and inequalities.

All participants in this study understood and were prepared to engage with our two core questions: *what factors drive child health and wellbeing in a local area* and w*hat factors drive health inequalities between different groups of children and young people in a local area*?. They recognised that there *is* a 'local system' that affects child health inequalities, and that a system map of influencing local factors on child health inequalities is meaningful. This suggests that there is value in taking a systems approach to the breadth of child health inequalities as a single public health issue, rather than tackling single issues such as obesity, healthy eating or physical activity.

Our methodological approach also has limitations. Group concept mapping workshops are time consuming and resource-intensive for busy local professionals. We also asked for their ongoing involvement and feedback on map development, to reach consensus and avoid researcher bias. While we provided feedback on the results of the local mapping workshops, we did not ask children, young people and carers to take part in the validation workshops and therefore they did not have the same opportunity as professional stakeholders to further refine the local maps. In addition, we undertook some consultation with CYP prior to starting the study but our core research questions were in large part generated by the academic research team, rather than by local communities or professional stakeholders, which may have limited their participation. The two local areas involved in the development of the map were selected to be diverse, to increase the applicability of the generic map in any English local area. We suggest that many of the factors on the map, and the domains into which they have been organised, are not unique to England and may have applicability in other countries though this is yet to be tested.

The map produced as a result of this study is complex and potentially overwhelming. Several participants noted that further work to develop a more interactive version of the map may

support easier interpretation and adaptation for use in local areas. It is qualitative, and does not demonstrate the relative strength of factors in their influence on child health inequalities. It also lacks information to understand the direction of influence between factors. It is not intended to be an accurate representation of an objective, observable system, but rather an accurate representation of the system that influences child health inequalities at the local level as understood by the expert participants involved in this study.

The potential utility of the map has been explored by respondents to our online survey stating that at the local level, senior leaders, policy makers, service commissioners and elected councillors would use the map as a reference tool for updating local policy, and to identify (and collect data) on factors that may be influenced by local public health interventions and hence inform evaluation design. The publication of the map and further dissemination will allow this to be fully explored further. Its value may lie in demonstrating to stakeholders the importance of policies that tackle the *systemic* drivers of child health inequalities beyond those traditionally associated with public health. The map provides a systems thinking framework for emerging Integrated Care Systems to consider the breadth of influences on child health inequalities and to reflect on their own role in the governance of that system.

Some participants commented that in the current climate of cuts to local authority budgets, and many local services contracting to deliver only statutory responsibilities, taking a systems approach to child health inequalities is not presently viable. The data collection for this study took place before the COVID-19 pandemic occurred, and its impact on the public health landscape in the UK is not yet fully understood. What is clear is that while the risk to children and young people of becoming seriously ill from the virus is very low, in the long-term child health inequalities will likely be exacerbated. Increased rates of financial instability will likely lead to a rise of child poverty in the UK, and children's charities are warning of the differential impact on vulnerable groups including children in care, asylum seeking children, and young carers [53]. The heightened impact of COVID-19 on those living in more deprived areas, and those in Black, Asian and Minority Ethnic (BAME) groups, means that children from these groups will be more likely to have been bereaved, or see their family income level drop [54]. The pandemic has been described as a systemic shock to the determinants of child health, and the immediate response to it has diverted resources away from child health and social care services, including acute services for life threatening illnesses, child protection services, and preventive services that support early years development [55]. The pandemic has forced public sector organisations to work in partnership and across the system. As we emerge from the immediate impact of COVID-19, there may be an opportunity for local systems to rethink how the system works, how services for CYP are delivered and health inequalities addressed; as local areas adjust to the 'new normal' they may also 'reset' and 'restore' service delivery to better support child health [56].

Those responsible for addressing child health outcomes may use the map to understand how the local system operates to produce inequalities, and identify potential points to intervene that may close the gaps [57], making the most of limited resources. Intervening in the local system may be in the form of a policy or intervention [58], but mapping the whole system also demonstrates how single programmes, policies or interventions are unlikely to impact on the complex and dynamic issue of child health inequalities. Childhood obesity studies have shown how some single setting-based approaches (e.g. schools, health services) show promise, but effects are not maintained once programmes end, and such programmes overlook the wider social and environmental determinants of obesity [50,59]. The Foresight obesity map has supported the hypothesis of solutions ranging from individual behaviour change through to policy settings, demonstrating the value of systems thinking [24]. However, a recent study exploring public health action on obesity in England found that, while local authorities

described the causes of obesity as complex systems, there was a mismatch in active programmes and policies which mostly focused on traditional individual-level interventions [60]. It is our hope that the map produced as a result of this study provides evidence that, for child health inequalities to be successfully addressed, coordinated action across a range of local authority, third sector, health, CCG and community groups is required.

## Supporting information

**S1 Appendix. Online survey details.**
(PDF)

**S2 Appendix. Mapping the child health system.**
(PDF)

## Acknowledgments

The views expressed are those of the authors and not necessarily those of the National Institute of Health Research or the Department of Health and Social Care.

The authors would like to thank Tarra Penney, Matt Egan, Harry Rutter and Elisabeth McGill for their advice on the methodology for this study.

## Author Contributions

**Conceptualization:** Patricia E. Jessiman, Katie Powell, Hannah Fairbrother, Ruth Kipping.

**Data curation:** Patricia E. Jessiman, Katie Powell, Philippa Williams, Hannah Fairbrother, Mary Crowder, Joanna G. Williams, Ruth Kipping.

**Formal analysis:** Patricia E. Jessiman, Katie Powell, Philippa Williams, Hannah Fairbrother, Mary Crowder, Joanna G. Williams, Ruth Kipping.

**Funding acquisition:** Patricia E. Jessiman, Katie Powell, Hannah Fairbrother, Ruth Kipping.

**Investigation:** Patricia E. Jessiman, Katie Powell, Philippa Williams, Hannah Fairbrother, Mary Crowder, Joanna G. Williams.

**Methodology:** Patricia E. Jessiman, Katie Powell, Philippa Williams, Hannah Fairbrother, Mary Crowder, Joanna G. Williams, Ruth Kipping.

**Project administration:** Patricia E. Jessiman, Katie Powell, Philippa Williams, Hannah Fairbrother, Mary Crowder, Ruth Kipping.

**Validation:** Patricia E. Jessiman, Katie Powell.

**Visualization:** Patricia E. Jessiman, Katie Powell.

**Writing – original draft:** Patricia E. Jessiman, Katie Powell, Philippa Williams, Hannah Fairbrother, Mary Crowder.

**Writing – review & editing:** Patricia E. Jessiman, Katie Powell, Philippa Williams, Hannah Fairbrother, Mary Crowder, Joanna G. Williams, Ruth Kipping.

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
