## [Decision Letter · Decision Letter 0]

13 Oct 2020

PONE-D-20-21649

A systems map of the determinants of child health inequalities in England at the local level

PLOS ONE

Dear Dr. Jessiman,

Thank you for submitting your manuscript to PLOS ONE. After careful consideration, we feel that it has merit but does not fully meet PLOS ONE’s publication criteria as it currently stands. Therefore, we invite you to submit a revised version of the manuscript that addresses the points raised during the review process.

Both reviewers comment that this is a very good paper and they raise a few minor suggestions prior to publication. Please could you address these in full and we look forward to reviewing your revised submission.

We look forward to receiving your revised manuscript.

Kind regards,

Fiona Cuthill, PhD

Academic Editor

PLOS ONE

Journal Requirements:

2. Please provide additional details regarding participant consent. In the ethics statement in the Methods and online submission information, please ensure that you have provided more details on the consent procedure in the ethics statement in the Methods and online submission information. As your study included minors, please state whether you obtained consent from parents or guardians.Moreover, please ensure that you have specified (1) whether consent was informed and (2) what type you obtained (for instance, written or verbal).

3. Please include additional information regarding the survey or questionnaire used in the study and ensure that you have provided sufficient details that others could replicate the analyses. For instance, if you developed a questionnaire as part of this study and it is not under a copyright more restrictive than CC-BY, please include a copy, in both the original language and English, as Supporting Information. Moreover, please include more details on how the questionnaire was pre-tested, and whether it was validated.

5. Please ensure that you refer to Figure 5 in your text as, if accepted, production will need this reference to link the reader to the figure.

Additional Editor Comments (if provided):

Reviewers' comments:

Reviewer's Responses to Questions

**Comments to the Author**

1. Is the manuscript technically sound, and do the data support the conclusions?

Reviewer #1: Yes

Reviewer #2: Yes

2. Has the statistical analysis been performed appropriately and rigorously? 

Reviewer #1: N/A

Reviewer #2: N/A

3. Have the authors made all data underlying the findings in their manuscript fully available?

Reviewer #1: No

Reviewer #2: Yes

4. Is the manuscript presented in an intelligible fashion and written in standard English?

Reviewer #1: Yes

Reviewer #2: Yes

5. Review Comments to the Author

Reviewer #1: This is a very interesting and well written article which uses an innovative qualitative systems approach to map the determinants of child health inequalities. The article currently provides a very thorough overview of the methodology and method, which takes the reader clearly to a full description of the interconnected domains of the map.

As it stands, the article broadly mets the seven criteria for acceptance to PLOS ONE for publication. However, I would recommend clarification and editing before submissions.

1. Goldfield et al's conceptual model of neighbourhood effects is relied upon for developing your own model. The article would benefit from some explanation as to why this conceptual model was most approporiate.

2. I would note that the various figures, unless I missed this, were not available to the reveiwer. The visual elements are likely essential to making the sections which describe the domains more engaging. Beyond the simple discription of the domains, the most important element is the intersectionality across domains. Will any of the figures illustrate these intersections?

3. The results repeats much of the methods already described (pages 16 -18). This may be a requirement of the journal, however, it felt unnecessary. Trimming would allow more space to discussion implications and contributions in more depth (see next point).

4. In the dicussion I agree that the method is the key strength in the study, and I can see how, at a local level this type of approach could prove incredibly insightful. While I don't disagree with the points made on the utility of the map, I felt more time could be spent elaborating and exemplifying these possible uses. The map is, as the authors note, complex, intersectional and qualitiative which introduces possible limitations to its use. Can the authors draw out how those responsible for addressing child health outcome might use the map in practice? What might such engagement / use look like? This is not to say I don't think this approach makes an important contribution - it does - but I think the authors would benefit from spending more time drawing this out. Perhaps including some reflective on next steps (where will you take the model next) might also be useful.

5. Finally an observation - and not something you can necessarily address here - but it was dissapointing to see that despite involving children and young people in developing the map, they were not consulted again at the later stages. Given this is a map about children and young people, and the issues directly affecting them, it would have been beneficial to see their own testimonies being included in the validation stage.

Reviewer #2: This is an excellent paper that outlines a systems mapping approach to understanding the factors that drive child health and inequalities in child health at a local area level in the UK.

The authors use a novel systems mapping approach informed by extensive qualitative interviews, policy review and stakeholder consultation in order to develop a systems map. They then do rigorous sense checking with policymakers, academics and practitioners to further refine the mapping. The end result is novel, I don’t think this is ever been done before, and I can see it making and I can see it making an important contribution to discussions in the area that I work, providing a comprehensive picture of the factors that one needs to consider when trying to address the complex challenges of reducing health inequalities in children.

The paper is very well written and easy to follow. I have just a couple of points for suggested improvements. I wonder if the authors could give some concrete examples of how this map might be used at a local area level, and adapted in order to inform or make changes to practice. Secondly it would be good to comment on the generalisability of the map and its application in other settings and countries – I actually think that many of the factors that they identify are universal, and that this map will travel well.

6. PLOS authors have the option to publish the peer review history of their article (what does this mean?). If published, this will include your full peer review and any attached files.

Reviewer #1: No

Reviewer #2: No

---

## [Author Response · Author response to Decision Letter 0]

26 Nov 2020

Response to reviewers,

Dear editor(s)

Thank you for the time taken to consider this article for publication, and to the peer reviewers for their thoughtful comments. We have responded to all queries as outlined below.

With warm wishes

Tricia Jessiman and co-authors

 Thank you, we have amended the formatting of the article and file names. All figures have been checked with PACE.

2. Please provide additional details regarding participant consent. In the ethics statement in the Methods and online submission information, please ensure that you have provided more details on the consent procedure in the ethics statement in the Methods and online submission information. As your study included minors, please state whether you obtained consent from parents or guardians. Moreover, please ensure that you have specified (1) whether consent was informed and (2) what type you obtained (for instance, written or verbal).

We have amended the ethics statement at the start of the methods section to include the information that “All participants were sent detailed information leaflets about the study, and had the opportunity to ask the research team questions about participation.” We have also added that “Where participants were under 16 years of age, parents/carers were also sent study information and had the opportunity to decline consent for their child to participate (the child also had to give written consent prior to participation).”

The statement also says that “All provided written informed consent prior to participation in the interviews, workshops (signed consent form), or online survey (online consent form).”

3. Please include additional information regarding the survey or questionnaire used in the study and ensure that you have provided sufficient details that others could replicate the analyses. For instance, if you developed a questionnaire as part of this study and it is not under a copyright more restrictive than CC-BY, please include a copy, in both the original language and English, as Supporting Information. Moreover, please include more details on how the questionnaire was pre-tested, and whether it was validated.

This was a qualitative survey and not validated, and we have added details about this:

 “The survey questions were developed by the research team and piloted with academic colleagues (the questions were qualitative).”

We have included details of both surveys as supplementary information (appendix 1)

Qualitative interview data (anonymised transcripts) for the scoping interviews used in this study has been deposited in ReShare. ReShare is the UK Data Service's online data repository, where researchers can archive, publish and share research data.

https://reshare.ukdataservice.ac.uk/

The data was deposited on 24.11.20 and is currently subject to review. It will not be published until it has been checked by the UK Data Service for disclosure risk, copyright breaches, etc.. The ReShare data collection number is 854532.

Once the checks have been conducted we will forward the DOI and accession details.

5. Please ensure that you refer to Figure 5 in your text as, if accepted, production will need this reference to link the reader to the figure.

 Thank you for spotting this omission, it has now been referred to in the text as follows “The governance domain (Figure 5)…”

Additional Editor Comments (if provided):

Reviewer #1: This is a very interesting and well written article which uses an innovative qualitative systems approach to map the determinants of child health inequalities. The article currently provides a very thorough overview of the methodology and method, which takes the reader clearly to a full description of the interconnected domains of the map.

As it stands, the article broadly mets the seven criteria for acceptance to PLOS ONE for publication. However, I would recommend clarification and editing before submissions.

We’d like to thank this reviewer for their positive and encouraging comments about the paper. We have responded to their recommendations in detail below, and in the paper.

1. Goldfield et al's conceptual model of neighbourhood effects is relied upon for developing your own model. The article would benefit from some explanation as to why this conceptual model was most approporiate.

Thank you. We have added the following explanation to the article: “Guided by a commitment to privileging our participants’ perspectives, we sought to ensure our analysis was data-driven (an inductive approach). However, following a review of existing frameworks of the determinants of child health inequalities and expert consultation, we used Goldfeld et al’s conceptual model of neighbourhood effects influencing early childhood development to organise the factors identified by participants and ‘clustered’ them into meaningful sub-systems or ‘domains’[41].Our approach therefore cohered with an appreciation that thematic analysis is ‘actively constructed’ by the researcher and that ‘analysis lays over bits of data to give them shape without doing violence to them’ (42). Goldfeld’s conceptual model offered the ‘best fit’ for working with our data as the domains echoed our data to a large extent.

2. I would note that the various figures, unless I missed this, were not available to the reveiwer. The visual elements are likely essential to making the sections which describe the domains more engaging. Beyond the simple discription of the domains, the most important element is the intersectionality across domains. Will any of the figures illustrate these intersections?

We are sorry that this reviewer did not see the domains and whole map. Yes, the whole map shows the intersectionality across domains in detail (Appendix 2)

3. The results repeats much of the methods already described (pages 16 -18). This may be a requirement of the journal, however, it felt unnecessary. Trimming would allow more space to discussion implications and contributions in more depth (see next point).

Thank you. We think we do need to give an account of the participants who contributed to the qualitative data collection, mapping workshops, and online survey as (as we point out in the discussion) we think it highlights the success of this methodology in engaging such a wide range of stakeholders.

4. In the dicussion I agree that the method is the key strength in the study, and I can see how, at a local level this type of approach could prove incredibly insightful. While I don't disagree with the points made on the utility of the map, I felt more time could be spent elaborating and exemplifying these possible uses. The map is, as the authors note, complex, intersectional and qualitiative which introduces possible limitations to its use. Can the authors draw out how those responsible for addressing child health outcome might use the map in practice? What might such engagement / use look like? This is not to say I don't think this approach makes an important contribution - it does - but I think the authors would benefit from spending more time drawing this out. Perhaps including some reflective on next steps (where will you take the model next) might also be useful.

Thank you. The focus of this paper is to describe a systems approach to mapping the determinants of child health inequalities at the local level in England for young people aged 0-25, and present and describe the resulting map. A second phase of the study will include further analysis of interviews about the utility of the map to stakeholders and system leaders, and will be the focus of a subsequent paper. 

We have edited the following paragraph in the discussion to expand further on the utility of the map:

“The potential utility of the map has been explored by respondents to our online survey stating that at the local level, senior leaders, policy makers, service commissioners and elected councillors would use the map as a reference tool for updating local policy, and to identify (and collect data) on factors that may be influenced by local public health interventions and hence inform evaluation design. The publication of the map and further dissemination will allow this to be fully eplored further. Its value may lie in demonstrating to stakeholders the importance of policies that tackle the systemic drivers of child health inequalities beyond those traditionally associated with public health. The map provides a systems thinking framework for emerging Integrated Care Systems to consider the breadth of influences on child health inequalities and to reflect on their own role in the governance of that system”. 

5. Finally an observation - and not something you can necessarily address here - but it was dissapointing to see that despite involving children and young people in developing the map, they were not consulted again at the later stages. Given this is a map about children and young people, and the issues directly affecting them, it would have been beneficial to see their own testimonies being included in the validation stage.

This is an important and valid point, and a criticism we accept. We did feedback to young people in both local site areas about the final results. In one area, CYP were consulted about changes to the map produced as a result of the CYP workshop, but consent was not taken to use the data from that session. We have acknowledged this limitation in the discussion section

“While we provided feedback on the results of the local mapping workshops, we did not ask children, young people and carers to take part in the validation workshops and therefore they did not have the same opportunity as professional stakeholders to further refine the local maps.”

Reviewer #2: This is an excellent paper that outlines a systems mapping approach to understanding the factors that drive child health and inequalities in child health at a local area level in the UK.

The authors use a novel systems mapping approach informed by extensive qualitative interviews, policy review and stakeholder consultation in order to develop a systems map. They then do rigorous sense checking with policymakers, academics and practitioners to further refine the mapping. The end result is novel, I don’t think this is ever been done before, and I can see it making and I can see it making an important contribution to discussions in the area that I work, providing a comprehensive picture of the factors that one needs to consider when trying to address the complex challenges of reducing health inequalities in children.

The paper is very well written and easy to follow. I have just a couple of points for suggested improvements. I wonder if the authors could give some concrete examples of how this map might be used at a local area level, and adapted in order to inform or make changes to practice. Secondly it would be good to comment on the generalisability of the map and its application in other settings and countries – I actually think that many of the factors that they identify are universal, and that this map will travel well.

Again, we would like to thank this reviewer for such positive feedback on our innovative approach. We have responded to the point about local area utility, as this was something that reviewer #1 also raised.

We think we need to be slightly cautious about international applicability. Of course, we hope that the map has value in other contexts and settings, but we have no evidence of this. We have added the following sentence to the conclusion of the paper.

“The two local areas involved in the development of the map were selected to be diverse, to increase the applicability of the generic map in any English local area. We suggest that many of the factors on the map, and the domains into which they have been organised, are not unique to England and may have applicability in other countries though this is yet to be tested.”

---

## [Decision Letter · Decision Letter 1]

5 Jan 2021

A systems map of the determinants of child health inequalities in England at the local level

PONE-D-20-21649R1

Dear Dr. Jessiman,

We’re pleased to inform you that your manuscript has been judged scientifically suitable for publication and will be formally accepted for publication once it meets all outstanding technical requirements.

Kind regards,

Fiona Cuthill, PhD

Academic Editor

PLOS ONE

Additional Editor Comments (optional):

Reviewers' comments:

Reviewer's Responses to Questions

**Comments to the Author**

1. If the authors have adequately addressed your comments raised in a previous round of review and you feel that this manuscript is now acceptable for publication, you may indicate that here to bypass the “Comments to the Author” section, enter your conflict of interest statement in the “Confidential to Editor” section, and submit your "Accept" recommendation.

Reviewer #1: All comments have been addressed

Reviewer #2: All comments have been addressed

2. Is the manuscript technically sound, and do the data support the conclusions?

Reviewer #1: Yes

Reviewer #2: Yes

3. Has the statistical analysis been performed appropriately and rigorously? 

Reviewer #1: Yes

Reviewer #2: N/A

4. Have the authors made all data underlying the findings in their manuscript fully available?

Reviewer #1: Yes

Reviewer #2: Yes

5. Is the manuscript presented in an intelligible fashion and written in standard English?

Reviewer #1: Yes

Reviewer #2: Yes

6. Review Comments to the Author

Reviewer #1: Many thanks for addressing my comments. I am happy to recommend that this article is accepted for publication. I look forward to seeing how the model develops in the future and its practical utility. Hopefully you will publish this as soon as you can as I think the contribution could be significant.

Reviewer #2: Thanks - the authors have done a good job of addressing all the peer review comments and I have no further comments

7. PLOS authors have the option to publish the peer review history of their article (what does this mean?). If published, this will include your full peer review and any attached files.

Reviewer #1: No

Reviewer #2: No

---

## [Editor Report · Acceptance letter]

4 Feb 2021

PONE-D-20-21649R1 

A systems map of the determinants of child health inequalities in England at the local level 

Dear Dr. Jessiman:

I'm pleased to inform you that your manuscript has been deemed suitable for publication in PLOS ONE. Congratulations! Your manuscript is now with our production department. 

Kind regards, 

on behalf of

Dr. Fiona Cuthill 

Academic Editor

PLOS ONE